

# Psychological stress responses to a live performance by professional flamenco dancers

Rosa de las Heras-Fernández[1], Guillermo Mendoza[2] and Manuel Jimenez[1]

[1] Universidad Internacional de La Rioja, Logroño, La Rioja, Spain
[2] Universidad de Málaga, Málaga, Andalucía, Spain

## ABSTRACT

Dance is a high demanding discipline that involves physiological and psychological pressures. The pressure increases when dancers perform in front of an audience that, on a physiological level, can generate hormonal responses similar to those of an athlete before a competition for social status. Low levels of testosterone (T) and high levels of cortisol (C) are related to a decrease in performance and an increase in the risk of injury. Therefore, this study sets out to analyse hormone response patterns in professional flamenco dance performances depending on whether the performances are completed successfully and whether there are differences by sex and professional category. Saliva specimens (2–5 ml) were taken from the participants before and after the performance. Samples were immunoassayed by duplicate to analyze momentary fluctuations in two hormones regularly used in studies with professional athletes. The results showed significant differences in solo dancers' T responses before and after the performance ($p < 0.01$), suggesting that the dancing role in the show (soloist or corps the ballet) and responsibility over the performance were important modulators to the hormone responses observed.

# INTRODUCTION

The performing arts, and dance, are characterised by demanding schedules that place significant pressure on psychological and physiological coping mechanisms during the performance season. Professional dancers have low trait anxiety (*Rojo, 2016*), and like most athletes, they resist psychological stress (*Gavrilyeva et al., 2016*). Nevertheless, stressful situations occur, and dancers and athletes have higher self-reported psychological stress and fatigue levels during performances or competition than in training periods (*Harrison et al., 2020*). Psychological stress in professional dance is a topic that has been studied by researchers, being considered in 14.8% of the research on dance in Korea between 1988 and 2019 (*Baik & Lee, 2019*). Specifically, high levels of psychological stress have been related to recurrent knee and cervical spine injuries in flamenco dancers (*Baena-Chicón et al., 2020*). Furthermore, *Amado et al. (2011)* explained that psychological pressure is high in dance and that the day of the performance becomes a context which generates stress and anxiety. Thus, it is evident that professional dancers need to use

Corresponding author
Rosa de las Heras-Fernández,
rosa.heras@unir.net

psychological abilities and effective coping strategies to fulfil improve their performance (*Coll, Mateos & De las Heras Fernández, 2021*). So, in dance, as occurs in sport, it is necessary to consider emotional and psychological aspects in order to be successful (*Machado, Ruaro & Geller, 2006*; *Spieler et al., 2007*; *Kruger, Elsunet & Potgieter, 2019*). It is essential to know what psychological abilities are the most important to develop (*Ruiz & Arruza, 2005*) and which coping strategies are effective to improving their performance (*Márquez, 2006*). Moreover, there are very few studies on professional dance that determine the differences in psychological stress with regard to gender (*Van Winden et al., 2020*), probably due to the low number of men who practice dance (*Ibáñez-Granados, Ortiz-Camacho & Baena-Extremera, 2018*; *Requena-Pérez, Martín-Cuadrado & Lago-Marín, 2015*). However, some studies done with female dancers found differences in psychological stress, showing greater stress in both dance students, due to the lack of time to be able to make it compatible with their studies (*García-Dantas & Caracuel, 2011*) and in professional dancers during their performance (*De Wet, Africa & Venter, 2022*). Other studies indicate that professional men dancers show psychological unease that is significantly more negative than the women dancers (*Hamilton et al., 1989*).

At a neuroendocrine level, stress is regulated by the hormone cortisol (C), which increases blood glucose levels, promotes tissue regeneration, and prepares the body for the fight or flight response. In sport, anticipatory physiological stress responses can be observed hours before a competition (*Elloumi et al., 2008*; *Díaz et al., 2013*), being more evident in men athletes, while in women athletes the anticipatory cortisol response has not been consistent (See: *Van Paridon et al., 2017* for a review). In dance, the anticipatory stress responses have been similar to those in sports, but not consistent. The cortisol levels associated with stress vary depending on the type of dance (*West et al., 2004*), the experience of the dancer (*Barcarolo et al., 2017*), and the presence of a competitive context (*Rohleder et al., 2007*).

Furthermore, in performances before an audience, dancers usually have high testosterone (T) levels, which are associated with euphoria, competitiveness, and self-esteem, age and training history also influence (*Łagowska & Kapczuk, 2016*). *Mazur (1985)* considered that high T levels favoured assertive facial expressions and interest in impacting others, while low T levels inhibited them. Testosterone fuels motivation to obtain status and achieve tangible incentives with which to improve one's hierarchical position within a social group (*Archer, 2006*; *Hall, Stanton & Schultheiss, 2010*; *Mazur, 1985*; *Mazur & Booth, 1998*; *Oliveira, 2004*; *Salvador, 2005*; *Van Anders & Watson, 2006*; *Mehta, Jones & Josephs, 2008*; *Stanton & Schultheiss, 2011*). Anticipatory T responses before a competition are higher in men than in women (*Kivlighan, Granger & Booth, 2005*). The most accepted explanation for the T–motivation association in humans is the biosocial model of status (*Archer, 2006*; *Hall, Stanton & Schultheiss, 2010*; *Mazur, 1985*; *Mazur & Booth, 1998*). Recent meta-analyses have suggested that in competitive situations, T is a hormone that regulates the body's systems (*i.e.*, physiological and psychological mechanisms), with its production increasing before an event. At the same time, we prepare ourselves to confront the challenge and increase even more abruptly when emotions of pleasure, happiness, or euphoria are experienced because of success (*Jiménez, Aguilar & Alvero-Cruz, 2012*;

*Geniole & Carré, 2019*; *Van Honk et al., 2004*). In agonistic encounters in nature and in uncertain situations where human beings are exposed to success or failure, psychophysiological responses are studied (*i.e.*, psychological, physiological, endocrine, and affective-emotional factors in response to success and failure). It suggests that mechanisms with a biological basis that react before competitive success (*Aguilar, Jiménez & Alvero-Cruz, 2013*; *Jiménez et al., 2020*; *Morgulev & Avugos, 2020*). Each person's tension in these life events could modulate momentary T fluctuations when challenges are faced with success. *Mazur (1985)* proposed that these increases in T provide behavioural feedback, acting as instrumental enhancers (*i.e.*, they are perceived as rewards or punishments), influencing how future challenges are confronted. Attaining success would reinforce the behavioural patterns that have enabled an individual to achieve it, in the medium and long term, encouraging a dominant and decisive style characteristic of people who have managed self-improvement (*Jiménez, Aguilar & Alvero-Cruz, 2012*; *Casto & Mehta, 2019*). However, few studies have been found that measure hormonal responses in dancers. *Murcia, Bongard & Kreutz (2009)* found a reduction in C and an increase in T in people dancing the tango in pairs, while *Łagowska & Kapczuk (2016)* suggest that high T levels are related to status within the dance academy in female ballet dancers. However, the samples analysed in both of these studies comprised amateur dancers.

For this reason, the objective of the present study was to determine whether these hormone response patterns are also present in professional dancers. Hormonal studies on competitive dancers have mainly focused on men with few investigations including both sexes (*Casto & Edwards, 2016*). Thus, we have taken a sample of professional men and women flamenco dancers when they face a performance before a live audience. We aim to establish whether performing before an audience modulates or not the production of androgens according to sex, as well as a comparison between solo dancers and the corps de ballet that accompany them.

## MATERIALS AND METHODS

The participants were an authoritative chosen sample of 14 professional dancers (10 women). Hence, it was not probabilistic. Participants were chosen only based on the principal researcher's knowledge of more than 20 years as a professional dancer. The selection process using judgmental sampling must be done carefully between active and highly professional quality dancers on tour. Bad performances may affect public attendance in subsequent shows and reduce their economic incentives. The sample was measured in three different shows, two at the "Taberna Pimkelton" and "Tablao Las Carboneras" in Madrid and one at "El Palacio del Flamenco" in Barcelona. Participants' saliva specimens were taken just one time in one of these shows, not three times per each.

The dance styles performed were Spanish and flamenco by six members of the ballet corps and flamenco by eight solo bailaores. A two-time longitudinal design was used. Saliva samples (5–10 ml) were taken in plastic tubes (Salivettes®, Barcelona, Spain) 40 min before and after the performance, between June and December 2018, with a mean interval between sampling of around 180 min. The decision was made to use only performances that took place in the evening to minimise the influence of circadian cycles on hormonal

fluctuations. The samples were taken between 18:30 h and 22:30 h. The *bailaores* were given instructions not to eat any food or brush their teeth for 30 min before the time of sampling. The samples at this stage were fully anonymised, frozen at −40 °C within 15 min and were conserved for analysis using an enzymatic immunoassay device (Grifols Triturus, Barcelona, Spain) at the Haematology Laboratory of the Hospital Universitario Virgen de la Victoria in Málaga. The samples were analysed in duplicate within the same test. The intra- and inter-assay coefficients of variation were (respectively) 6.5% and 8.7% for testosterone and 8.3% and 11.9% for cortisol. The lower detection limits for the testosterone and cortisol kits were 3.5 pg/ml and 0.05 ng/ml (Diametra, Milan, Italy) respectively. Finally, the power effect for the listed interactions $(1 - \beta)$ was analysed detecting more than 90% of type II error. Participants received a document explaining the benefits and costs of participating in the study, reading and signing the informed consent. The experiment was approved by the Ethics Committee of the Universidad de Malaga (Spain) with registration number: CEUMA-35-2018-H.

## Statistical analyses

The Shapiro-Wilk test of normality was carried out, and it was observed that the hormonal variables were not distributed normally. To normalise them, they were log transformed, giving a normal distribution (the figures are presented with raw values so that they can be compared with earlier studies). A repeated measures $2 \times 2$ ANOVA was applied to analyse the concentrations of T × Group (soloists and corps de ballet) and C × Group, as well as the T × Sex and T × Group × Sex interactions. The effect size was also calculated with the value of $\eta^2_p$. The Mann-Whitney U test for non-parametric data was used to analyse the differences between groups, and the non-parametric Wilcoxon test was used to analyse the intragroup differences. The percentage changes in T during the performance were calculated following the formula used by *Jiménez, Aguilar & Alvero-Cruz (2012)* to study hormonal fluctuations: $[(\text{post} - \text{pre}) \times 100/\text{pre}]$. All statistical analyses were conducted using the statistical package SPSS® 20 (IBM Co, Armonk, NY, USA).

## RESULTS

The sample comprised 14 professional dancers aged between 23 and 48 years, of whom four were men and 10 women. T presented significant intrasubject differences in the T × Group interaction ($f_{(1,10)} = 39.46$, $p = 0.00009$, $\eta^2_p = 0.80$), and for the T × Sex interaction ($f_{(1,10)} = 11.70$, $p = 0.007$, $\eta^2_p = 0.54$) (Fig. 1). No differences were observed in the T × Sex × Group interaction, or in C in any of the interactions studied (Fig. 2). The intersubject differences were only observable in the T × Sex interaction ($f_{(1,10)} = 6.16$, $p = 0.03$, $\eta^2_p = 0.38$). Differences by sex were confirmed by the test for independent samples, with a higher production observed in men (Z = −2.69, $p = 0.007$). The same test was used to analyse the differences according to responsibility in the performance. A test for related samples was performed and neuroendocrine responses between the start and the end of the performance were observed in the soloists but not the corps de ballet (Z = −2.52, $p = 0.012$). The percentage differences in the T responses before and after the performance

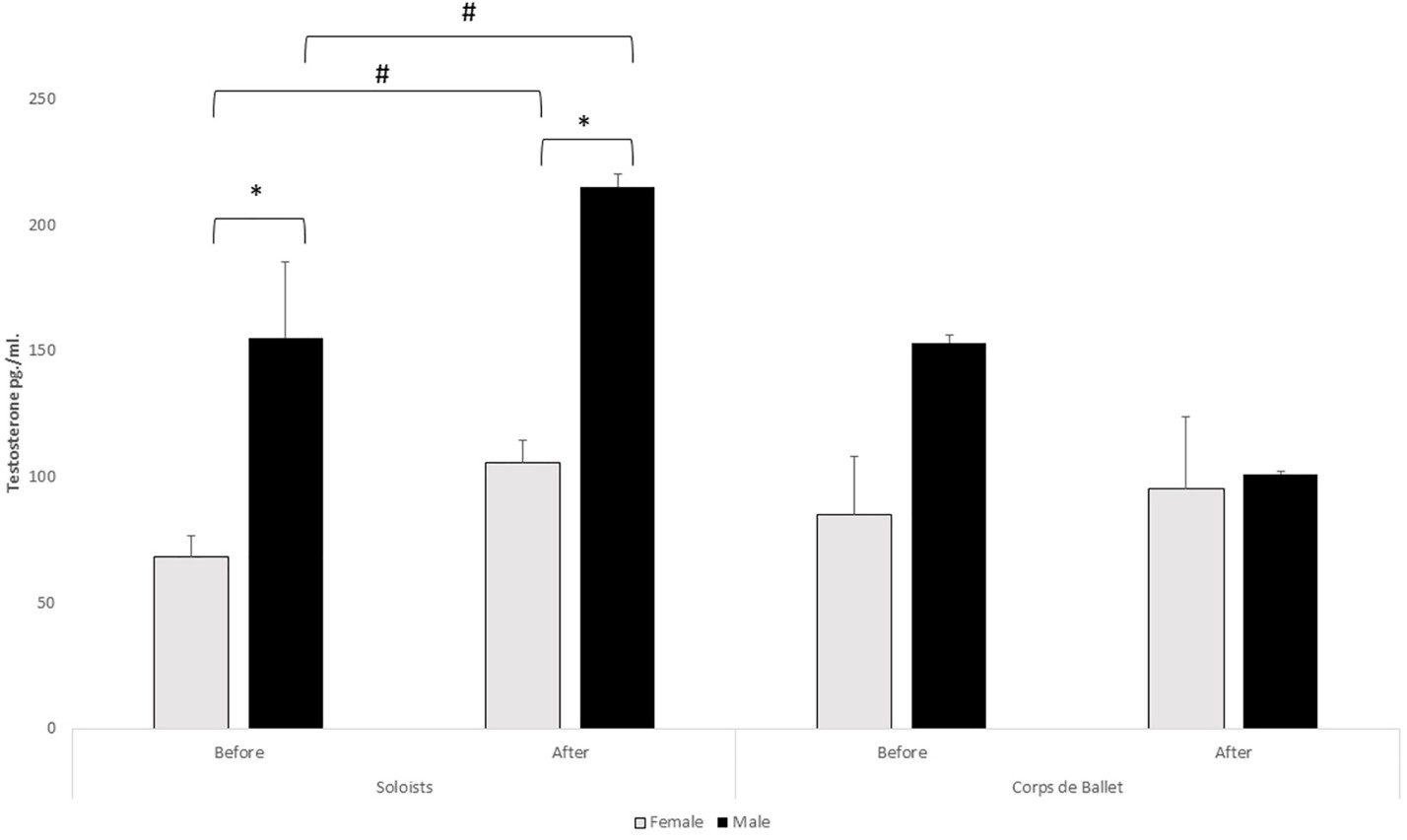

**Figure 1 Testosterone levels by sex and between solo dancers and corps de ballet.** *Differences between sexes ($p < 0.01$); #intragroup differences ($p < 0.01$). The error bars represent the standard error of the mean.

were (Mean ± SE): 56.11 ± 9.71% for the solo *bailaores* and −5.05 ± 9.47% for the corps de ballet, showing significant differences between groups (Z = −2.97, $p = 0.003$).

## DISCUSSION

We aim to establish whether performing before an audience would modulate or not the production of androgens according to sex, as well as a comparison between solo dancers and the corps de ballet that accompany them. The study shows higher testosterone production in men. It is necessary to remember that male dance could be another indicator of masculine competitiveness (*Hugill et al., 2009*). The highest testosterone levels in men agree with other studies in which men displayed higher objectives concerning dancing professionally and working as a soloist and devoting themselves to choreography (*Beleña, 2017*).

The slight increase of testosterone in the dance corps de ballet is also observed in works relating to sport, which find that testosterone does not change after friendly games. In these games, the players did not perceive a real threat of dominance (*Jiménez et al., 2020*). In the same way, the dance corps de ballet do not feel a psychological threat when their performance is collective and collaborative through joint actions (*Reddish, Fischer & Bulbulia, 2013*). For this reason, the corps de ballet presents less psychological concern and

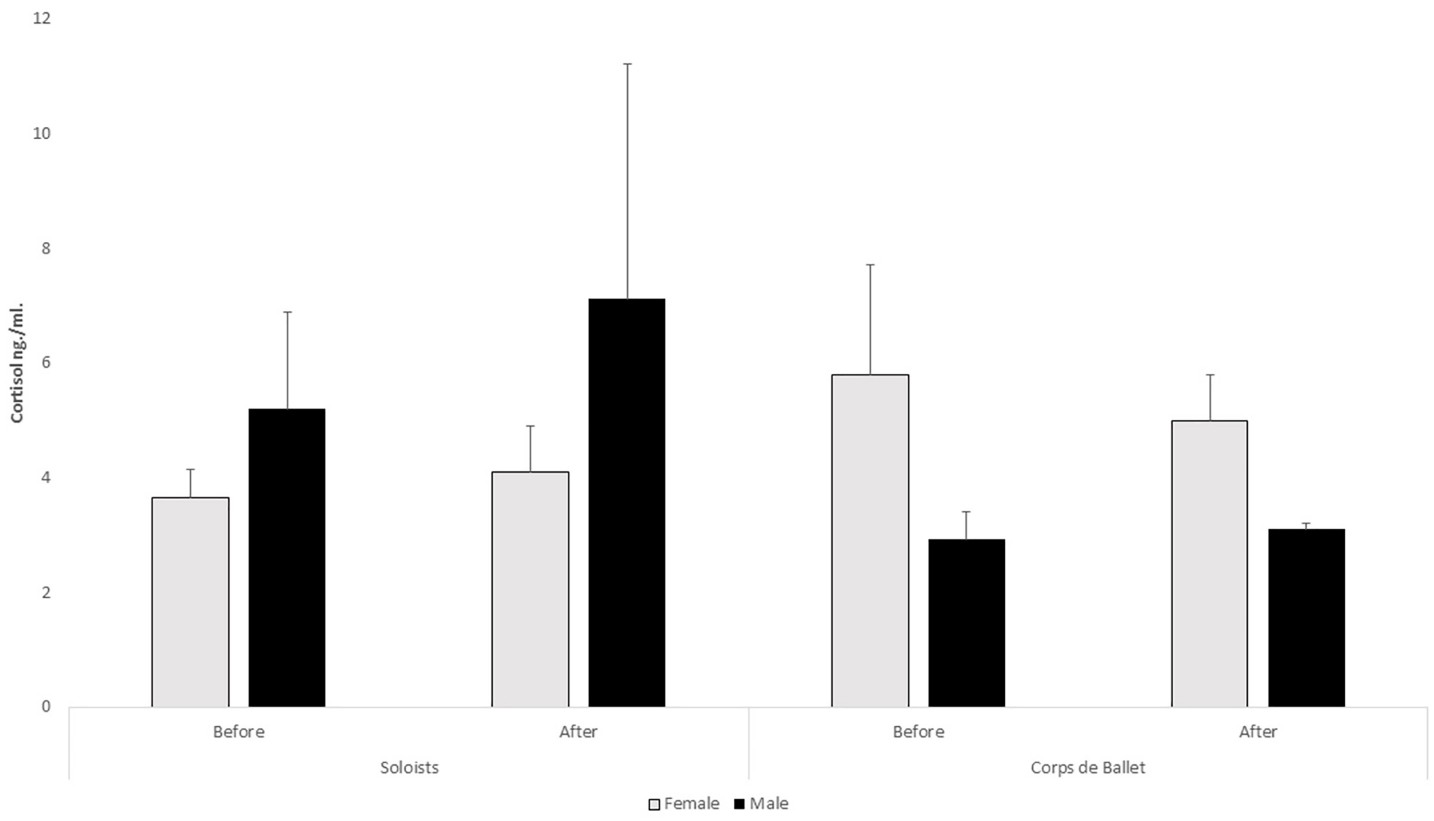

**Figure 2 Cortisol levels by sex between solo dancers and the corps de ballet.** The error bars represent the standard error of the mean.

trainability than soloists/principal dancers (*De las Heras-Fernández et al., 2020*), and lower scores in anxiety, are found in the dancers that "accompany", that is, only intervene at some moments and do not have an essential role in the show, and therefore face less risks and responsibility (*Coll, Mateos & De las Heras Fernández, 2021*). On the other hand, the high T levels of professional flamenco soloist dancers match results based on status in amateur ballet (*Łagowska & Kapczuk, 2016*).

In the saliva cortisol test, no significant stress increase was found in the soloists or the corps de ballet, in contrast to what happens in similar studies with this test in ballroom dancing competitions (*Strahler & Luft, 2019*). Most likely a competitive setting, where a winner and loser is proclaimed, becomes a factor that modulates stress as already was observed in competitive dancing (*Rohleder et al., 2007*). The results of physiological stress suggest that in flamenco dance, the dancer's role (solo or in the corps de ballet) does not influence the levels of stress shown during a performance. However, recent studies have indicated that the corps of ballet scored lower in psychological stress and anxiety than the soloist/principal dancers, suggesting less concern in public shows (*Coll, Mateos & De las Heras Fernández, 2021*). In other subjects from the "scenic arts" like the professional orchestra musicians who were studied psychological stress after the final performance in the International Festival of Bergen, increases were found in performances with an audience regardless of their "role" (soloists or the rest of the members of the orchestra)

(*Halleland et al., 2009*). After the research was carried out, it can be seen that there is little research on physiological stress responses in dance, which has led to citing studies on dance that refer to psychological stress.

Given that any dance performance before an audience generates stress levels, applying the psychological coping strategies used in sports would be interesting. The theoretical framework by *Lazarus & Folkman (1986)* proposes a model based on controlling emotions, organising the input of information, planning the responses and executing the appropriate actions. Thus, abilities like concentration, confidence, self-motivation, commitment, simulation or self-regulation become essential when explaining success. Furthermore, psychological aptitudes are more critical the more significant the competitive demand (*Cantón, Checa & Ortín, 2009*; *Cruz, 1995*). On the other hand, it would be interesting to implement an intervention programme in dance where the subjects could improve their levels of the different indicators of mental health like stress, anxiety, self-esteem, and self-confidence (*Moledo & López, 2013*). Dance, movement, and therapy programmes have been implemented in professional dancers, decreasing their stress levels (*Jung Eun & Kyung Soon, 2020*). Through creative dance, positive effects have been achieved on emotional abilities (*De Rueda & López, 2013*). Moreover, the strategies aimed at reducing stress could potentially reduce the incidence of injuries in dance, as the physical aspects seem to be related to the psychological ones, and injuries are associated with a complex combination of biopsychosocial factors (*Cahalan et al., 2015*). Thus, improving psychological aspects could foment the development of dance in the different agents in the sector: dancers, teachers, schools and professional companies (*Van Winden et al., 2021*).

## CONCLUSIONS

The results of this study lead us to think that beyond the venue where the dancer gave their performance, the role they played may be a determinant for their T responses. Soloists have more exposure and more pressure for their performance, and it is coherent that they showed a peak in T after a successful show. Regarding stress, the null results obtained due to the variability in C levels among dancers suggest that other variables could play as a stress modulator beyond the role played in the show.

### Limitations

Regarding the study's limitations, firstly, after measuring the physiological stress responses in the flamenco dancers after the performance, this study cannot determine to what extent the stressor of performing in front of a live audience caused a psychological stress response. In addition, we are still determining if the observed responses could result from the live performance with an audience since the medical performances without an audience have yet to be repeated. On the other hand, this study was carried out in tablaos where there are smaller and usually less specialized audiences so that they present a minimal level of exposure, risk and critical repercussions (public or journalists) that can affect the maintenance of the performances, and therefore, a situation involving less stress. For this, other studies could be carried out in other stage spaces with a more significant influx of the public and/or a more specialized public. For future studies, it would be interesting, in

addition to measuring physiological responses, to measure psychological responses to triangulate data and obtain better results. In addition, the size of the sample should be highlighted. It must be taken into account that there are far fewer professional dancers than elite athletes. In any case, it would be desirable to conduct this study with a higher significant sample of dancers, considering different venues, dance styles and professional categories.

### Funding
The authors received no funding for this work.

### Competing Interests
The authors declare that they have no competing interests.

### Author Contributions
- Rosa de las Heras-Fernández conceived and designed the experiments, performed the experiments, authored or reviewed drafts of the article, and approved the final draft.
- Guillermo Mendoza conceived and designed the experiments, performed the experiments, analyzed the data, prepared figures and/or tables, authored or reviewed drafts of the article, and approved the final draft.
- Manuel Jimenez performed the experiments, analyzed the data, prepared figures and/or tables, and approved the final draft.

### Human Ethics
The following information was supplied relating to ethical approvals (*i.e.*, approving body and any reference numbers):

The experiment was approved by the Ethics Committee of the Universidad de Malaga (Spain) with registration number: CEUMA-35-2018-H.

### Data Availability
The raw measurements are available as a Supplemental File.

### Supplemental Information
Supplemental information for this article can be found online at http://dx.doi.org/10.7717/peerj.15282#supplemental-information.

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
