# Peer review of "Psychological stress responses to a live performance by professional flamenco dancers"

_PeerJ, doi:10.7717/peerj.15282_

## Round 0.1 · original submission · Major Revisions

Some changes are needed to the article.

Georgian Badicu
Academic Editor
PeerJ Life & Environment

Reviewer 1 ·

Basic reporting

This is a very interesting article, but the focus of both the Introduction and Discussion sections of the article tend to stray away from what was studied. The overarching topic is stress, where there is a stressor (e.g., an event or situation) that, in turn, in psychologically interpreted by the brain to be a "stressful" situation, which then causes a physiological response (i.e., change in C and/or T). In this study, only the physiological responses are measured (i.e., C and T), but there is huge amount of energy spent on talking about the psychology of stress (e.g., causes, coping strategies, etc). Even the title emphasizes "Psychological" as the first word where, in fact, there are no psychological evaluations of stress whatsoever. The title also implies that there is a longitudinal component to this study (i.e., "data for performer training and development", and the authors do, in fact, talk about this issue in the Discussion, but it is not studied. For example, the authors could have used some like the Profile of Mood States (POMS) questionnaire to assess acute changes in mood as a reflection of stressors. A better title for this study could have been some like "Physiological stress responses to a live performance by professional flamenco dancers". To be honest, the current title does not make any sense relative to the content of the current article. Throughout the manuscript, the word "stress" is often used, but without clarification about what kind of stress is being talked about - i.e., psychological or physiological.

Abstract - The authors should reformat the content of the abstract to adhere to common abstract formatting guidelines which includes the use of an introduction (currently too long), a purpose statement, a methods sections (currently missing), results section, and a conclusion statement (currently confusing).

Introduction - If the essential topic of this paper is to evaluation how and whether C and T change (pre versus post) acutely in response to a stressor (i.e., participating in a show), then the Introduction should mainly focus on preparing the reader to understand that topic. Currently, the authors have spent a little too much time talking about the psychological interpretation of stress, but I do see that as the authors' choice to use this section in that manner.

Discussion - The first sentence and/or paragraph should remind the reader about the purpose of the study before launching into an interpretation of the results. Study limitations should include the fact that the authors do not, in fact, know to what degree that the stressor (performing in front of a live audience) actually caused a psychological stress response (which then would have caused the physiological stress responses of elevated C and T).

Conclusions - There is no conclusions section.

Experimental design

The first sentence of the Methods section describes the participants of this study, but this, in fact, is results. What should be described here instead is the target population of interest and whether there were specific inclusion and/or exclusion criteria for recruitment. So, information about sample sizes by gender and ages should be relocated to the start of the Results section.

It is clear that the saliva samples were collected around the dancers' participation in three different flamenco shows, but were samples collected for every dancer for every show or was some other measurement strategy in place. This is just not clear to from the information presented.

Validity of the findings

Limitations not stated include... The authors do not, in fact, know to what degree that the stressor (performing in front of a live audience) actually caused a psychological stress response (which then would have caused the physiological stress responses of elevated C and T). In addition, there is no control measures such as repeating the same measures on the same dancers but to an empty auditorium. Thus, we don't really know whether the observed responses are due to live performance or not - It is assumed.

There is no conclusions section for this paper.

Additional comments

There are numerous wording and grammatical errors that can be corrected. I'll highlight some that caught my eyes, but please do not interpret this to mean that this is a complete and exhaustive list of these errors.

Abstract
* Lines 22-24 - Suggest rewording this sentence.
* Lines 31-32 - "suggesting that the levels of exposure and responsibility and the impact of the performance are important" - This does not make any sense as written.
* Lines 32-33 - This conclusion statement does not make any sense relative to the rest of the abstract.

Introduction
* The word "stress" is used throughout this section without usually no reference to the kind of stress (psychological or physiological). Should review the entire section and try to edit accordingly.
* Line 46 - Use "schedules" instead of "calendars".
* Lines 51-52 - Delete "as performing before an audience produces greater tension.
* Lines 54-55 - Reword to something like "Specifically, high levels of stress have been related to recurrent injuries in the knee and cervical spine in flamenco dancers".
* Line 64 - "satisfy a dancer's expectations..."
* Lines 67-70 - Should reword this entire sentence as I cannot quite understand the point being made.
* Line 81 - Insert comma after "(Barcarolo, et al., 2017)".
* Line 84 - Replace "and their" with "with".
* Line 85 - Replace "have an influence" with "having an influence".
* Line 87 - Replace "T" with "Testosterone" as abbreviations should generally not be used to start a sentence.
* Line 90 - "&amp" - is this correct?
* Line 100 - "experimented"? Do you mean "studied"?
* Line 110 - Remove parentheses from around "Murcia et al. (2009)"
* Line 112 - Remove comma after "Kapczuk"
* Line 112 - "related to status in women" - what status?
* Line 113 - Change to "in both of these studies"
* Line 114 - Change "analyze" to "determine"
* Lines 115-116 - Change to "Hormonal studies on competitive dancers have mostly focused on men with few investigations having included both sexes..."
* Lines 118-119 - "We aim to establish whether performances ending with success (i.e., the audience attends the event, and it is completed successfully)..." I really don't understand what this statement is trying to say except that the performance was in front of a live audience?
* Line 120 - Change to "as well as a comparison between solo dancers..."

Methods
Line 124 - In addition to moving this sentence as suggested previously, also change to "23 and 48 years"
Line 136 - I'm going to assume that this should be changed to read as "-40 C within 15 minutes"
Line 141 - Change to "0.05 ng/ml (Diametra, Milan, Italy), respectively.

Statistical Analyses
* Lines 155-156 - Delete this last sentence as it is a repeat of lines 143-144 in the previous section.

Results
Both Figures - The wording for the legends of both figures should be edited. Specifically, both note that the figures are showing differences when, in fact, they are showing means and standard error bars.

Discussion & Conclusions
* Lines 197-202 - Run-on sentence. Should separate this into at least two separate sentences reword accordingly.
* Lines 204-207 - The wording in these sentences is difficult to understand.
* Lines 222-226 - Run-on sentence. Should separate this into at least two separate sentences reword accordingly.
* Line 232 - Add comma after "movement"
* Lines 232-234 - Suggest rewording this sentence as it is difficult to understand.
* Lines 241-244 - Suggest adding other relevant limitations as described earlier.

Reviewer 2 ·

Basic reporting

The authors report a useful, carefully designed and conducted study of professional male and female flamenco dancers (soloists and corps) with respect to measures of testosterone and cortisol before and after performances for live audiences. The straightforward findings are analysed, interpreted, and discussed smartly, with appropriate scholarly sophistication. The paper will be useful to several kind of research specialities, and doubtless will move the areas of research forward sensibly.

Experimental design

No comment, see above

Validity of the findings

No comment, see above

---

## Round 0.2 · Minor Revisions

Some Minor Revisions.are required.

With kind regards,
Georgian Badicu
Academic Editor
PeerJ Life & Environment

Reviewer 1 ·

Basic reporting

The authors have done a great job of revising this manuscript. I have read everything again and came up with a short list of minor issues that the authors should consider. I realize that between this review and the last review that I have made repeated suggestions about adding commas to lists within a sentence. These suggestions are based upon my own preference for comma usage which, in turn, are based upon the Oxford Comma rule.

Experimental design

No additional comments

Validity of the findings

No additional comments

Additional comments

Abstract
Line 14 – Change start of sentence to read as “The pressure increases” (no capital P and an “s” for increase).
Line 16 – “social status searching..” – First, remove the second period at end of sentence. Second, I think that the word “searching” is incorrect somehow – i.e., either it’s the wrong word or it should be deleted.
Line 27 – remove second period at end of sentence.
Introduction
Line 64 – Insert a comma after “regeneration”.
Line 74 – Insert a comma after “competitiveness”.
Line 80 – Remove second semi-colon following “2005”.
Line 87 – Remove comma after “challenge”.
Line 88 – Insert comma after “happiness”.
Line 92 – Insert comma after “endocrine”.
Methods
Line 135 – Add a space between “within” and “15”.
Results
I saw that the original description of the dancers was removed from the Methods section, but I don’t see a replacement for this description in the Results section. I would suggest a 1-2 sentence description at the very beginning of the Results section to tell the readers about your research participants – i.e., basic demographics for each sex and sample size for each.
Discussion
Line 185 – Change this to read as “… an audience would modulate …”
Line 195 – Add a space between “threat” and “of”.
Line 209 – Change to read as “The results of physiological stress”
Line 212-214 – Suggest rewording this sentence.
Line 219 – Change “ther” to “there”.
Conclusions
Line 245 – Change to read as “the role they played may be a determinant for…”
Limitation
Line 252 – Replace “After” with “after”.
Line 257 – Replace “This” with “this”.

---

## Round 0.3 · Minor Revisions

Some minor changes are needed:

Keywords: it is good that some of these keywords to not be found in the title of the article, i.e., stress, performance, professional dance.

Statistical analyses: what was the statistical program used in this study and which version? please fill with the necessary information.


With kind regards,

Georgian Badicu
Academic Editor
PeerJ Life & Environment

---

## Round 0.4 · accepted · Accept

Thank you for your submission to PeerJ. Two reviewers recommend this paper to be published it. I congratulate you and your team for such great achievement.